# Silencing P2X7R Alleviates Diabetic Neuropathic Pain Involving TRPV1 via PKCε/P38MAPK/NF-κB Signaling Pathway in Rats

**DOI:** 10.3390/ijms232214141

**Published:** 2022-11-16

**Authors:** Lisha Chen, Hongji Wang, Juping Xing, Xiangchao Shi, Huan Huang, Jiabao Huang, Changshui Xu

**Affiliations:** 1Department of Physiology, Basic Medical College of Nanchang University, Nanchang 330006, China; 2Jiangxi Provincial Key Laboratory of Autonomic Nervous Function and Disease, Nanchang 330006, China; 3The Clinical Medical School, Jiangxi Medical College, Shangrao 334000, China; 4The First Affiliated Hospital, Jiangxi Medical College, Shangrao 334000, China

**Keywords:** P2X7 receptor, TRPV1, DNP, neuron, satellite glial cell

## Abstract

Transient receptor potential vanillic acid 1 (TRPV1) is an ion channel activated by heat and inflammatory factors involved in the development of various types of pain. The P2X7 receptor is in the P2X family and is associated with pain mediated by satellite glial cells. There might be some connection between the P2X7 receptor and TRPV1 in neuropathic pain in diabetic rats. A type 2 diabetic neuropathic pain rat model was induced using high glucose and high-fat diet for 4 weeks and low-dose streptozocin (35 mg/kg) intraperitoneal injection to destroy islet B cells. Male Sprague Dawley rats were administrated by intrathecal injection of P2X7 shRNA and p38 inhibitor, and we recorded abnormal mechanical and thermal pain and nociceptive hyperalgesia. One week later, the dorsal root ganglia from the L4-L6 segment of the spinal cord were harvested for subsequent experiments. We measured pro-inflammatory cytokines, examined the relationship between TRPV1 on neurons and P2X7 receptor on satellite glial cells by measuring protein and transcription levels of P2X7 receptor and TRPV1, and measured protein expression in the PKCε/P38 MAPK/NF-κB signaling pathway after intrathecal injection. P2X7 shRNA and p38 inhibitor relieved hyperalgesia in diabetic neuropathic pain rats and modulated inflammatory factors in vivo. P2X7 shRNA and P38 inhibitors significantly reduced TRPV1 expression by downregulating the PKCε/P38 MAPK/NF-κB signaling pathway and inflammatory factors in dorsal root ganglia. Intrathecal injection of P2X7 shRNA alleviates nociceptive reactions in rats with diabetic neuropathic pain involving TRPV1 via PKCε/P38 MAPK/NF-κB signaling pathway.

## 1. Introduction

More than 500 million people worldwide have diabetes, and the prevalence of diabetes is increasing worldwide [1]. Moreover, China has a substantial problem with type 2 diabetes [2]. Diabetic neuropathy is the most severe complication of diabetes and is associated with nerve pain, sensory disturbances, increased mortality rate, and amputation [3]. The clinical manifestations of peripheral neuropathy in diabetic patients are symmetrical pain and paresthesia, which are more common in the lower extremities than in the upper extremities, and seriously reduce the quality of life of most diabetic patients. Unfortunately, there are no internationally recommended drugs or suitable protocols for treating diabetic neuropathic pain (DNP) [4]. Therefore, there is an urgent need to understand the pathogenesis and pathogenesis of diabetic neuropathy. So, it is necessary to establish the model of DNP to study its mechanism by intraperitoneal injection of streptozocin (STZ) [5] and measurement of pain-stimulated response [6].

As we know, the P2X7 receptor (P2X7R) is a cation channel activated by a high concentration of adenosine triphosphate (ATP) and is closely associated with various diseases [7]. The P2X7R in L4-L6 dorsal root ganglion (DRG) contributes to diabetes-induced nociceptive sensitization, which prevents DNP [8] and decreases thermal nociceptive sensitization [9]. P2X7R is highly expressed in satellite glial cells (SGCs) and is involved in inflammatory responses [10]. Transient receptor potential vanillic acid 1 (TRPV1), which is involved in neurogenic inflammatory processes [11], is sensitive to irritant vanillic acid compounds, heat, low pH, and inflammatory factors [12]. So, the contribution of TRPV1 in pain has been widely reported in recent years as a detector of painful stimuli produced by chemicals and heat. The TRPV1 cation channel is expressed at a high level in neurons of the DRG, and DNP is driven by the activation of TRPV1 ion channels [13]. On the contrary, inhibition of the TRPV1 channel of the DRG could reduce diabetes-mediated pain [14]. In addition, inhibition of P2X7 receptor expression and inflammatory response can reduce thermal sensitization and prevent diabetic neuropathic pain [15]. Because both TRPV1 and P2X7 have been implicated in inflammatory pain, the relationship between them has attracted much attention. P2X7R has been found to have a good association with TRPV1 in esophageal inflammation because of their closed correlated gene expression [16]. Meanwhile, long et al. [17] reported that neuropathic pain is related to the expression of P2X7R and TRPV1. Injection of P2X7 inhibitors can reduce TRPV1 expression associated with diabetic neuropathic pain [8].

Studies have shown that protein kinase C (PKC), p38 mitogen-activated protein kinase (P38 MAPK), and nuclear factor-κB (NF-κB) are thought to regulate TRPV1 activity [18,19]. TRPV1 is downstream of PKC signaling and is associated with pain [20,21]. Inhibition of PKCε and TRPV1 expression in the peripheral nervous system can prevent the transition from acute to chronic pain [20]. P38 MAPK is phosphorylated by PKC signaling and can also regulate the expression of TRPV1 [19]. P38 MAPK is a MAPK pathway involved in signaling cascades and can be activated and phosphorylated by various stimuli to participate in neuropathic pain-related cellular responses [22]. NF-κB is a major transcription factor associated with inflammatory responses and pain [23]. Inhibition of the p38 MAPK/NF-κB pathway can alleviate the physiological activities of neuropathic pain [24]. Moreover, inhibition of the NF-κB signaling pathway by recombinant lentiviral vectors significantly inhibited TRPV1 expression [25]. Therefore, PKCε/P38 MAPK/NF-κB is expected to bridge the interaction between P2X7 and TRPV1.

We found that P2X7 shRNA and p38 inhibitor significantly reduced the expression of TRPV1, PKCε, pp38, and pp65, expression of inflammatory factors in DNP rats. Therefore, this study aimed to determine whether the PKCε/p38 MAPK/NF-κB signaling pathway participates in diabetic neuropathic pain involving P2X7R and TRPV1 in rats.

## 2. Results

### 2.1. Establishment of a DNP Model

MWT and TWL in the model rats were lower than in the normal rats (*p* < 0.01; Figure 1A,B). Within one week of STZ injection, the model rats’ thresholds of thermal and mechanical pain were significantly lower, and significant plantar pain and thermal hyperalgesia appeared (*p* < 0.01). Two weeks later, all rats in the modeling rats developed hyperalgesia (*p* < 0.01), suggesting that the rats had been modeled from the perspective of behavior.

### 2.2. Nerve Conduction Velocity

Nerve damage retards conduction. To determine whether there is neurological impairment in type 2 DNP rats, we measured sciatic nerve conduction velocities. The DNP rats had a slower electrical conduction velocity than the Control rats (*p* < 0.01; Figure 2), suggesting that DNP is a type of nerve damage. There were no significant differences among the NC shRNA, DMSO, and DNP groups (*p* > 0.05); the P2X7 shRNA and p38 inhibitor groups had faster conduction velocity than the DNP group (*p* < 0.01), suggesting that silencing P2X7R and using p38 inhibitor ameliorates the injury.

### 2.3. Blocking of P2X7R/P38 MAPK May Relieve Hyperalgesia

To understand the effect of P2X7shRNA and p38 inhibitor on rat behavior, we measured thermal and mechanical pain in all rats. At the end of the sixth week, all the rats in the modeling group had nociceptive sensitivity. In subsequent studies, only rats in the model group in which STZ treatment increased fasting blood glucose above 7.8 nM were used. At the end of the seventh week, MWT and TWL were higher in the P2X7 shRNA group than in the DNP group (*p* < 0.01; Figure 3A,B), and these numerical behavioral values were higher in the p38 inhibitor groups than in DNP rats (*p* < 0.01; Figure 3A,B). There were no significant differences among the NC shRNA, DMSO, and DNP groups (*p* > 0.05). P2X7 shRNA reduced mechanical and thermal nociceptive sensitization in the DNP rats, suggesting that DNP in rats could be alleviated by silencing P2X7R. The p38 inhibitor also reduced injury sensitization of DNP rats, suggesting that DNP in rats can be alleviated with inhibition of the p38 MAPK signaling pathway.

### 2.4. The P2X7R on SGCs

The qPCR assay, Western blotting, and immunofluorescence analysis of P2X7R expression were performed in rats. The level of P2X7 mRNA in each group was detected by qPCR, and the results show that the level of P2X7R in the DRG of DNP rats was significantly higher than in the Control group (*p* < 0.01; Figure 4A). In order to obtain quantitative information on the changes of P2X7R in diabetic neuropathic pain rats, the expression of P2X7R protein in rat DRG was studied by western blotting (Figure 4B,C). The western blotting revealed no difference between the NC shRNA and the DNP groups (*p* > 0.05). The expression of P2X7R in DRG of DNP rats was significantly higher than in the Control group (*p* < 0.01), while the expression level of P2X7R in the P2X7 shRNA group was lower than in the DNP group (*p* < 0.01). These results suggest that P2X7R is involved in DNP.

The co-expression of P2X7R with GFAP, an SGC marker, was examined using immunofluorescence double-labeling (Figure 4D). Activated satellite glia have been linked to pain [8]. We quantified the expression of GFAP in each group (Figure 4E). The fluorescence intensity of GFAP in the DNP group was more robust than that in the Control group (*p* < 0.01), while the fluorescence intensity of GFAP in the P2X7 shRNA group was lower than that in the DNP group (*p* < 0.01). These results indicated that the DNP group’s satellite glial cell activation was increased, and P2X7shRNA could prevent this activation. There was no difference in co-expression of the P2X7R and GFAP in Satellite glial cells of DRG between the NC shRNA and the DNP groups (*p* > 0.05). Co-expression in the DNP group was lower than in the Control group (*p* < 0.01; Figure 4F), while the P2X7 shRNA group showed lower co-expression than the DNP group (*p* < 0.01; Figure 4F). These findings suggest that P2X7 shRNA inhibits the co-expression of GFAP and P2X7R in DNP rats.

### 2.5. Blocking of P2X7R/P38 MAPK Downregulates TRPV1 on Neuron

We performed a qPCR assay, western blotting, and immunofluorescence analysis of TRPV1 expression. The qPCR assay (Figure 5A) and Western blotting (Figure 5B,C) revealed no differences among the NC shRNA, DMSO, and DNP groups (*p* > 0.05); P2X7 shRNA and p38 inhibitor groups showed lower expression than in the DNP group; DNP group showed higher expression of TRPV1 than in the Control group (*p* < 0.01), indicating that P2X7 shRNA and P38 inhibitor reduce TRPV1 levels in DNP rats.

The presence of co-expression of TRPV1 with NeuN (a neuronal marker) was examined using immunofluorescence double-labeling (Figure 5D). The dorsal root nerve TRPV1 receptor was co-expressed with NeuN. The P2X7 shRNA and p38 inhibitor groups showed lower co-expression levels than the DNP group, DNP group showed higher co-expression than the Control group (*p* < 0.01; Figure 5E), suggesting that the P2X7 shRNA and p38 inhibitor reduced the co-expression of NeuN and TRPV1 in DNP rats.

### 2.6. Measurement of TNF-α, IL-1β, PKCε, P38 MAPK and NF-κB

We measured the expression of the TNF-α in the P2X7 shRNA group, and p38 inhibitor groups were lower than in the DNP group (*p* < 0.01; Figure 6A,B), while the expression level of IL-1β in the P2X7 shRNA group and p38 inhibitor groups was lower than in the DNP group (*p* < 0.01; Figure 6C,D).

We measured proteins from DRGs tissues to measure the expression of the PKCε, P38 MAPK, and NF-κB in vivo (Figure 6E). We can see from Figure 6F that the expression of PKCε in the P2X7 shRNA group and p38 inhibitor groups was lower than in the DNP group (*p* < 0.01). The P2X7 shRNA and p38 inhibitor groups showed significantly lower levels of pp38 protein than the DNP group (*p* < 0.01; Figure 6G). Similarly, the protein levels of pp65 detected in the P2X7 shRNA and p38 inhibitor groups were lower than those in the DNP group (*p* < 0.01; Figure 6H).

## 3. Discussion

Neuropathic pain is a common disease caused by lesions or diseases of the central or peripheral somatosensory nervous system [26], and chronic neuropathic pain is characterized by nociceptive hypersensitivity, nociceptive abnormalities, altered sensation, and spontaneous pain [27]. The most common complication of diabetes is peripheral neuropathy which manifests as chronic neuropathic pain. However, the pathogenesis of DNP is not completely clear, and the treatment is difficult, which seriously disturbs diabetic patients. Therefore, a diabetic neuropathic pain rat model was successfully established in this study by measuring thermal, mechanical pain and sciatic nerve conduction velocity in rats (Figure 1 and Figure 2) to study its pathogenesis. Many studies have reported that both TRPV1 and P2X7 are involved in the formation of DNP [15,28], and some have reported that they are related between P2X7 and TRPV1 [8,17]. However, how they are related is still unknown, and our study is designed to fill that gap and provide a basis for future treatment. In this study, we explored the effects of P2X7 shRNA and p38 inhibitors on the development of DNP, and our results illustrated that inflammatory factors released upon activation of P2X7R located on SGC act on neurons and that inhibition of P2X7R expression reduces TRPV1 expression and relieves DNP via PKCε/P38 MAPK/NF-κB signaling pathway. These findings provide a basis for studying pathways for DNP pathogenesis and suggest targets for DNP therapy, and these findings also provide evidence for the communication between SGCs and neurons.

TRPV1, the first discovered member of the TRPV family (TRPV1-6), is the most widely studied channel protein, which high expression in specific afferent sensory fibers such as unmyelinated C fibers, thin myelinated Aδ fibers, and ganglion vagal neurons. When TRPV1 is activated by protons, heat, capsaicin or sensitized to inflammatory factors and second messengers [29], it generates electrical signals as the first step in the pain mechanism [30]. TRPV1 then regulates the release of cytokine release and neuropeptides to regulate pain [31]. It is well known that TRPV1 is one of the targets of inflammatory factors, including TNF-α and 1L-1β. Knockdown of the 1L-1R gene in highly TRPV1-expressing mice alleviated mechanical pain [32], while pretreatment with TRPV1 inhibitor in subcutaneously injected 1L-1β rats significantly alleviated heat pain allergy [33]. Both agonists [34] and inhibitors [35] of TRPV1 affect TNF-α-induced pain behavior, and TNF-α inhibitors can modulate neuropathic pain by reversing the up-regulation of TRPV1 [36]. Moreover, relief of peripheral neuropathic pain occurs by inhibiting TRPV1 upregulation in DRG and glial cell activation and TNF-α expression [37]. The behavioral analysis of our study demonstrated that the DNP group had lower thresholds for thermal and mechanical pain than the Control rats (Figure 1), while the protein blot and qPCR results further demonstrated that the TRPV1 protein expression in the DNP group was higher than the normal rats (Figure 5). These results suggest that activation of TRPV1 ion channels drives DNP [13]. The P2X7 receptor is a cation channel activated by high concentrations of adenosine triphosphate [7]. It is well known that P2X7R in the L4-L6 DRGs contributes to diabetes-induced hyperalgesia, and mice injected with intrathecal P2X7R inhibitors, and P2X7R knockout mice have significantly reduced pain sensation [38]. Our results after intrathecal injection of P2X7 shRNA also confirmed the above view. P2X7R expression was lower in the P2X7 shRNA group than in DNP rats (Figure 4), and P2X7 shRNA was found to reduce mechanical hyperalgesia (Figure 3) and improve the delay in sciatic nerve conduction velocity (Figure 2). Compared with DNP rats, the expression of TRPV1 was lower in the P2X7 shRNA group (Figure 5), indicating that inhibition of P2X7R would reduce the expression of TRPV1 and achieve a therapeutic effect. Therefore, our results suggest that P2X7R is involved in TRPV1-mediated DNP.

We have established that P2X7R is involved in TRPV1-mediated diabetic neuropathic pain, so how is P2X7R involved in this process? Several studies have shown that PKCε, P38 MAPK and NF-κB signaling pathways are involved in TRPV1-mediated neuropathic pain. [8,18,19]. It is well known that lipid-activated PKCε has been considered an essential mediator of intolerance and hepatic insulin resistance [39]. PKCε in major afferent nociceptors can induce a switch in intracellular signaling pathways to mediate cytokine-induced nociceptor activation, which may be a neuronal switching mechanism responsible for the functional transition from acute to chronic pain states [20]. PKCε is highly expressed in nociceptive neurons as a second messenger [40]. Moreover, its activity enhanced the sensitivity of the TRPV1 receptor to capsaicin [41]. Inhibition of the PKCε/TRPV1 signaling pathway can reduce neuropathic pain [42]. P38 MAPK is activated in neurons and glial cells and plays an essential role in inflammation and neuropathic pain [43]. Extracellular stimuli activate p38 MAPK through a series of responses [44]. Activated p38 MAPK regulates gene transcription and contributes to peripheral and central sensitization [45]. Inhibition of the P38 MAPK/NF-κB signaling pathway reduces cytokine release [46]. The most critical subunit in the NF-κB signaling pathway is p65, and the Ser534 phosphate of mouse p65 is not essential for its nuclear translocation [47]. As we know, prolonged activation of P2X7R increases the release of inflammatory factors and the inflow of calcium ions into the intracellular environment [7]. The increase of inflammatory factors further activates glial cells to release more such as inflammatory factors, ATP or glutamate via the P2X7 pathway, forming a cascade amplification effect. As shown in Figure 6, the expression of TNF-α and 1L-1β in the P2X7 shRNA group was lower than that in the DNP group, indicating that TNF-α and 1L-1β played a role in the DNP mediated by P2X7 and TRPV1. TNFα [48] and 1L-1β [49] can enhance calcium influx and increase neuronal activity, and PKCε is a protein kinase activated by calcium and diacylglycerol, which activates the mitogen-activated protein kinase (MAPK) cascade [50]. We found that PKCε, pp38 and pp65 were reduced in the P2X7 shRNA group and p38 inhibitor group compared with the DNP group, indicating that PKCε, pp38 and pp65 play a role in the DNP mediated by P2X7 and TRPV1. Compared with the DNP group, the protein and mRNA expression of TRPV1 were decreased in the P38 inhibitor group, while the expression of PKC was lower in the P2X7 shRNA group and the p38 inhibitor group. These results suggest that P2X7R regulates the expression of PKCε upstream of TRPV1 and that P2X7R participates in DNP through the PKC/p38 MAPK pathway. NF-κB is activated in the DRG of diabetic rats and co-locates with TRPV1 in most of the same neurons [25]. The expression of pp65 in the P2X7 shRNA group and p38 inhibitor group was lower than that in the DNP group, and the expression of TRPV1, pp38 and pp65 in the P2X7 shRNA group and p38 inhibitor group was lower, suggesting that PKCε/P38 MAPK/NF-κB was playing a role in DNP involving P2X7 and TRPV1. Our results have shown that inflammatory cytokines are mediators of the association between P2X7 located in satellite glial cells and TRPV1 located in neurons. Inflammatory cytokines can affect neurons [19] and glial cells and also lead to the activation of intracellular signaling pathways [48]. It has been demonstrated that neurotransmitters act on neurons and affect the PKC/p38/p65 pathway [19,46]. Moreover, the PKCε/P38 MAPK/NF-κB pathway could also be active in glial cells [50]. Our experimental results show that the content of inflammatory factors and PKC, pp38 and pp65, is also changed after the injection of P2X7 shRNA and p38 inhibitor in DRG, but the specific mechanism of action still needs further study. Therefore, we will further explore the specific mechanism at the cellular level in the future.

## 4. Methods

### 4.1. Animal Models and Experimental Groups

Changsha Tianqin Biological Co., Ltd. provided the experimental animals. The ethics committee of Nanchang University approved the procedures. For the male Sprague Dawley rats that were just bought, we acclimated them for 2 h, and then we did behavioral measurements (Time is denoted as 0), and rats were fed with conventional chow for 1 week (at the end of the first week, time is denoted as 1) and then fed with homemade high sugar and high -fat chow for 4 weeks; and at the end of this event (at the end of the fifth week, time is denoted as 1), STZ (S8050; Solarbio, Beijing, China) was injected intraperitoneally at 35 mg/kg after fasting for more than 12 h. The rats were successfully modeled as having DNP by measuring fasting blood glucose ≥ 7.8 mmol/L for 12 h or random blood glucose ≥11.1 mmol/L by tail vein blood collection with significantly increased behavioral sensitivity to pain. The successful rat models were randomly divided into DNP group, DNP plus intrathecal injection of P2X7 shRNA group (P2X7 shRNA group), DNP plus intrathecal injection of NC shRNA group (NC shRNA group; NC means negative control, NC shRNA is the shRNA of the scrambled sequence), DNP plus intrathecal injection of p38 inhibitor SB203580 (p38 inhibitor group; SB203580 is an inhibitor of p38 MAPK activity), and DNP plus intrathecal injection of solvent DMSO (DMSO group, the solvent was saline with 1% DMSO), with six rats in each group. Except for the DNP group, all successfully modeled rats were administrated by intrathecal injection of the corresponding drug at the end of the sixth week (time is denoted as 6). We performed behavioral tests at the end of weeks 0, 1, 5, 6, and 7. Six healthy male rats were taken as the Control group (fed with regular chow from the beginning to the end). Rats were anesthetized with 10% chloral hydrate after 12 h of fasting, and we measured nerve sciatic conduction velocity. The DRGs of the lumbar 4–6 nerves were removed and placed in RNA preservation and fixation solutions at −80 °C and −4 °C for subsequent experiments.

### 4.2. Intrathecal Injection

We injected 10% (30 mg/kg) chloral hydrate intraperitoneally with volume fraction. One hand located the most prominent L5 spinous process space, and the other used a 20-µL micro-syringe to puncture the lumbar vertebra clearance with an evident sense of breakthrough. P2X7 shRNA and NC shRNA were dissolved in the transfer reagent proportional to the rat’s body weight, and the final volume was 20 µL. The Control and DNP groups were not processed, and the P2X7 shRNA group rats were administrated by intrathecal injection of P2X7 shRNA, while the NC shRNA group rats were injected with NC shRNA at the end of the sixth week. SB203580 (HY-10256; MedChemExpress, Monmouth Junction, NJ, USA) was dissolved in saline with 1% DMSO (10 µg/µL; D8371; Solarbio, Beijing, China), 10 µL volume of the mixture was injected intrathecally in the P38 inhibitor group rats, and 10 µL volume of the solvent was injected intrathecally in the DMSO group at the end of the sixth week.

### 4.3. Behavioral Testing

An electronic, mechanical pain meter (BME-404; Tianjin, China) and automatic thermal pain stimulator (BME-410C; Tianjin, China) were used to measure the mechanical withdrawal threshold and Thermal withdrawal latency.

(1)Mechanical withdrawal threshold (MWT)

Injurious stress stimuli were used to assess mechanical pain sensitivity. Untethered rats were placed in mesh cages on layers of stainless steel wire mesh. After reaching a stable resting state, the plantar aspect of the foot on a wire mesh (1 cm × 1 cm) was stimulated using von Frey filaments, and the stress data were transmitted to the computer to measure foot retraction responses to mechanical stimulation. Each rat was measured six times at 10-s intervals, and all measurements were performed in a randomized manner.

(2)Thermal withdrawal latency (TWL)

Heat-sensitive pain was assessed by injurious thermal stimulation using a plantar stimulation system. Rats were placed the panel in a clear, bottomless plastic case on a glass. A light beam penetrating the glass plate was directed at the sole to produce thermal radiation stimulation. The beam was turned off when the rat’s paw was removed, and the time from the start of beam irradiation until the rat’s paw was removed was recorded. This period was designated the heat-shrinking foot latency period. Each rat was measured six times, alternating every 5 min on the side of the right paw. The maximum duration of thermal stimulation was 30 s. All measurements were performed in a randomized manner.

### 4.4. Neurophysiological Measurements

Nervous conduction velocity was recorded using a multi-channel physiological signal acquisition system type RM6240B (Chengdu Instrument Factory, Chengdu, China). The anesthetized limbs were fixed on a board with the rat’s back up. After exposing the right side of the sciatic nerve, stimulating electrodes were placed in the muscles surrounding the sciatic nerve. The recording electrodes connecting channels 1 and 2 were placed sequentially near the stimulating electrodes along the sciatic nerve without touching each other. The measurement parameters were as follows: stimulation intensity, two times threshold: wave width, 2–5 µs; single stimulation. The distance between electrodes 1 and 2 was recorded. The computer automatically recorded the conduction time between electrodes 1 and 2. The ratio of the distance of the two recording electrodes (m) and their conduction time (s) is the sciatic nerve conduction velocity (m/s). The measurement was repeated three times, and the average value was recorded.

### 4.5. Quantitative Real-Time Reverse Transcription Polymerase Chain Reaction (qPCR)

RNA was extracted using an Eastep^®^ Super Total RNA extraction kit (Promega, Beijing, China). The RNA lysates and ganglia were ground on ice in a ribozyme-free homogenizer according to the instructions, and the mixture was transferred to a nuclease-free Eppendorf tube after grinding for 10 min. All centrifugations were carried out using a fourth-degree centrifuge (Eppendorf, Hamburg, Germany). The tubes were then centrifuged at 14,000× *g* for 5 min; 0.5 times the volume of absolute ethanol was added to the extracted supernatants; mixed, and transferred to a centrifugation column, which was centrifuged at 14,000× *g* for 1 min; and washed with RNA wash solution for centrifugation at 14,000× *g* for 1 min. Then, 50 µL DNAase incubation solution was added to the center of the centrifugation column and was kept for 15 min, followed by two washes with 600 µL RNA solution at 14,000× *g* for 1 min. Finally, RNA was obtained by adding 30 μL nuclease-free water to the center of the centrifugation column membrane, standing for 2 min, and centrifuging for 1 min. EasyScript First-Strand cDNA Synthesis SuperMix (TransGen Biotech, Beijing, China) was chosen to synthesize complementary DNA from extracted RNA via reverse transcription at 42 °C. Complementary DNA was amplified using the PerfectStart Green qPCR SuperMix (TransGen Biotech, Beijing, China) in MyGene Series Thermal Cycler (LongGene, Hangzhou, China). CFX96^TM^ Real-Time System (Bio-Rad, Hercules, CA, USA) measured P2X7 and TRPV1 mRNA expression levels. The primers were used according to a previous report [8]. The reaction conditions were as follows: 94 °C for 30 s activation; 94 °C for 5 s; and 60 °C for 30 s amplification, a total of 45 cycles. The dissolution curve temperatures were as follows: 95 °C for 15 s; 60 °C for 1 min; 95 °C for the 30 s; 60 °C for 15 s. The internal reference gene CT values normalized the CT values of the target gene, and ΔCT was calculated; ∆∆CT = ∆CT test sample − ∆CT calibrator sample; finally, the expression level relative to the Control group was calculated as relative quantity = 2^−ΔΔCT^.

### 4.6. Western Blotting

The expression of TRPV1, P2X7R, IL-1β, and TNF-α and the PKCε/p38 MAPK/NF-κB signaling pathway was measured using western blotting. DRGs tissues from each group were collected and placed in a homogenizer, and pre-chilled radioimmunoprecipitation assay buffer (R0010; Solarbio, Beijing, China) was mixed with protease and phosphatase inhibitors proportionally. The lysates were mechanically ground until homogenates without visible clumps were obtained and incubated on ice for 30 min. The homogenate was centrifuged at 14,000× *g* for 10 min at 4 °C, and the supernatant was transferred to a new centrifuge tube while its volume was measured. We added an appropriate amount of protein loading buffer (P51114; TransGen Biotech, Beijing, China) and boiled the mixture sample for 5 min. SDS-PAGE isolated the protein, transferred it to immuno-Blot PVDF membranes (Solarbio), then blocked with 5% skim milk powder (LP0031B; Solarbio, Beijing, China) and incubated with primary antibody. P2X7 (APR-008, Alomone, Jerusalem, Israel), 1:800; TRPV1 (Bioss, Beijing, China) 1:1000; p38 (Cell Signaling Technology; CST, Boston, MA, USA), 1:800; phosphor- p38 (CST, Boston, USA), 1:800; PKCε (Proteintech, Hubei, China), 1:800; p65 (CST, Boston, USA), 1:2000; pp65 (CST, Boston, USA), 1:800; IL-1β (Affinity Biosciences, Jiangsu, China), 1:300; TNF-α (Boster, Wuhan, China), 1:500, were to incubate PVDF membranes overnight at 4 °C. Anti-β-actin (Zhong Shan-Gold Bridge, Beijing, China) was used to incubate membranes at a ratio of 1:800 for 2 h on ice with slow shaking. Finally, the corresponding secondary antibody was incubated at a ratio of 1:2000 for 1–2 h incubation on ice with slow shaking after membrane washing. The membrane was observed using ChemiDoc^TM^ XRS+ (Bio-Rad, California, USA) after the hypersensitive ECL luminescent solution (FD8020; Fude Biotechnology Co., Ltd., Hangzhou, China) was dropped. The optical densities of the target bands were measured using Image J software (National Institutes of Health, Bethesda, ML, USA). The relative expression levels of TRPV1, P2X7, PKCε, IL-1β, TNF-α, p38 MAPK, and NF-κB were calculated using the corresponding β-actin band as a control.

### 4.7. Immunofluorescence

After heart perfusion, the L4-L6 DRGs and spinal cord tissues of corresponding segments were placed in Eppendorf tubes and preserved with 4% paraformaldehyde. After 2 h of dehydration with 30% sucrose, DRG tissues were embedded with OCT, and 7–8 μm thick sections were prepared by freezing section mechanism. The sections were removed from the −20 °C refrigerator and washed with phosphate-buffered saline (PBS; P1010; Solarbio, Beijing, China) buffer, then re-fixed with a drop of 4% paraformaldehyde. After 10 min, the tissues were washed three times with 1 × PBS for 5 min each and then perforated with 0.3% Triton-X-100 penetration (T8200; Solarbio, Beijing, China). After 15 min of penetration, the cleaned and dried slides were blocked with 10% mountain donkey serum (SL050; Solarbio, Beijing, China) to reduce non-specific binding, and then the corresponding antibody was added to the tissue and incubated overnight. Primary antibodies were: P2X7R (1:100, rabbit); pp38 (1:100, rabbit), TRPV1 (1:100, rabbit), pp65 (1:100, rabbit), NeuN (1:200, mouse), and GFAP (1:200, mouse). Sections were incubated with secondary antibodies at 37 °C for 1 h: TRITC-conjugated goat anti-rabbit (Zhong Shan-Gold Bridge, 1:200) and FITC-conjugated goat anti-mouse (Zhong Shan-Gold Bridge, 1:200). Finally, the picks were placed under an inverted fluorescent microscope for imaging (Olympus, Tokyo, Japan). Since individual glial cells are sometimes difficult to distinguish, colocalization fluorescence intensity analysis was performed using Image-Pro Plus (Media Cybernetics Inc., Rockville, MD, USA) in co-labeling experiments, and GFAP expression was quantified in DRG satellite glial cells.

### 4.8. Statistical Analysis

IBM SPSS Statistics 21 software (IBM, Chicago, IL, USA) was used for statistical data analysis, and GraphPad Prism 6.01 (GraphPad Software Inc., San Diego, CA, USA) was used for image rendering. One-way analysis of variance was combined with the least significant difference test and the Tamhane4T2 test for differences between groups of normally distributed data. Results were expressed as mean ± standard error of the mean, with *p* < 0.05 or *p* < 0.01 indicating a statistically significant difference.

## 5. Conclusions

P2X7 shRNA and P38 inhibitors attenuate injury perception and reduce TRPV1 upregulation, and inhibition of P2X7 receptors reduces IL-1β and TNF-α release, suggesting that P2X7 receptors on glial cells are involved in DNP involving TRPV1 via inflammatory factors, and silencing P2X7R alleviates DNP involving TRPV1 via PKCε/P38 MAPK/NF-κB pathway.

## Figures and Tables

**Figure 1 ijms-23-14141-f001:**
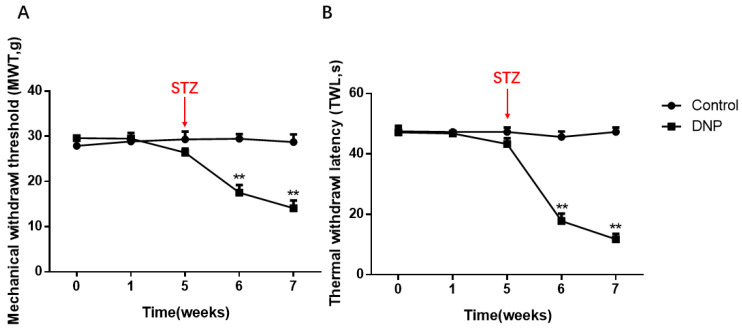
Mechanical withdrawal threshold (MWT) and Thermal withdrawal latency (TWL) in the DNP and Control groups. (**A**) Changes of MWT in normal and model rats. (**B**) TWL changes in the normal and model rats. The MWT and TWLs values of the DNP rats were significantly lower than those of the normal rats. The X-axis means the time of measuring behavior and drug administration, the detailed descriptions of 0, 1, 5, 6 and 7 are given in the Section 4, and we injected the rats intraperitoneally with streptozocin (STZ) at the end of the fifth week; data are shown as mean ± SEM based on at least three independent experiments, *n* = 6; ** *p* < 0.01 versus the normal rats.

**Figure 2 ijms-23-14141-f002:**
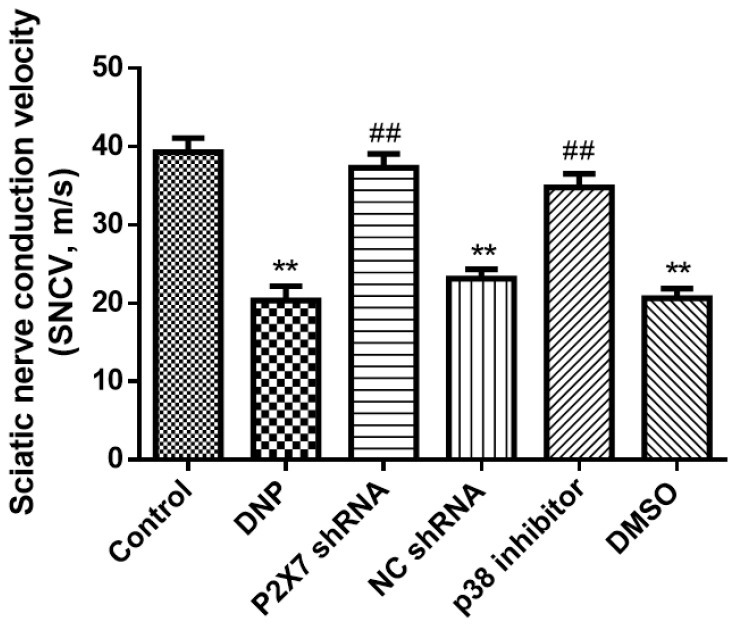
Sciatic nerve conduction velocity. The nerve conduction velocity of the DNP group was significantly lower than that of normal rats; however, there were no differences among the DNP, P2X7 shRNA, and NC groups. Intrathecal injection of P2X7 shRNA and p38 inhibitors decreased the rate of nerve injury. Data are shown as mean ± SEM based on at least three independent experiments, *n* = 6; ** *p* < 0.01 versus the Control group; ## *p* < 0.01 versus the DNP group.

**Figure 3 ijms-23-14141-f003:**
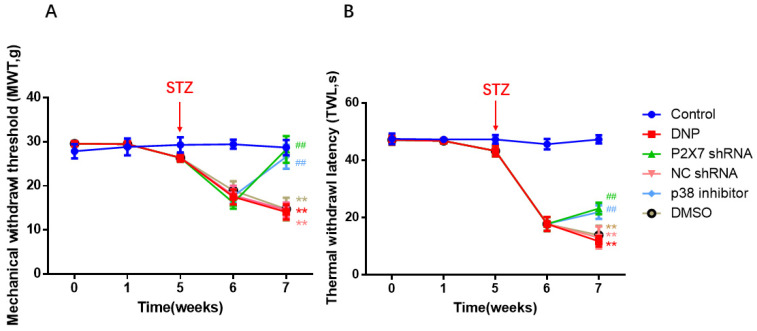
Results of MWT and TWL of all groups. (**A**) Changes in MWT values. (**B**) Changes in TWL values. At the end of the seventh week, MWT and TWL in the P2X7 shRNA and p38 inhibitor groups were higher than in the DNP group. The X-axis means the time of measuring behavior and drug administration, the detailed descriptions of 0, 1, 5, 6 and 7 are given in the Section 4; data are shown as mean ± SEM based on at least three independent experiments, *n* = 6; ** *p* < 0.01 versus the Control group; ## *p* < 0.01 versus the DNP group.

**Figure 4 ijms-23-14141-f004:**
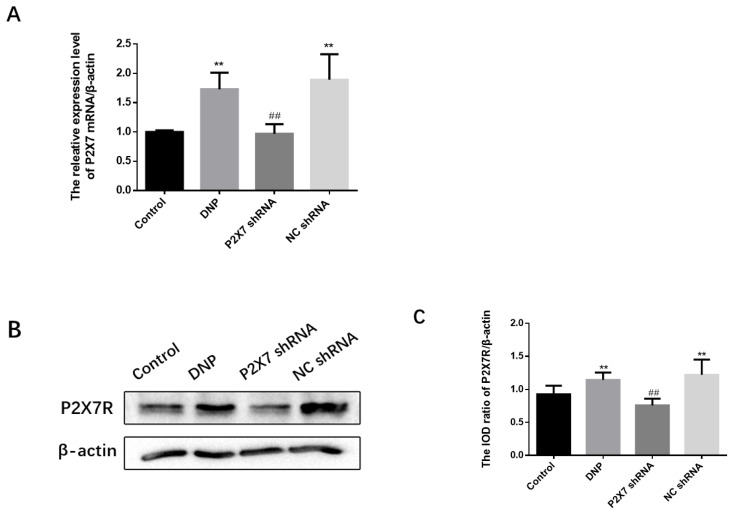
Measurement of P2X7 mRNA and protein levels in the DRG and co-expression levels of P2X7R with GFAP. (**A**) P2X7 mRNA. (**B**) SDS-PAGE of P2X7R protein in DRG of experimental rats in each group. (**C**) P2X7R protein expression in the DRG of each group was analyzed. (**D**) Co-expression of GFAP and P2X7R in DRG. (**E**) Relative fluorescence intensity levels of GFAP in each group. (**F**) Relative fluorescence intensity levels of GFAP and P2X7R co-expression. P2X7 mRNA, protein levels, and co-expression levels of P2X7R and GFAP in the P2X7 shRNA group were lower than in the DNP rats. Blue staining (DAPI) marks the nucleus, green staining (GFAP) marks activated SGCs, and red staining marks P2X7R. The scale bar represents 50 µm; data are shown as mean ± SEM based on at least three independent experiments, *n* = 6; ** *p* < 0.01 versus the Control group; ## *p* < 0.01 versus the DNP group.

**Figure 5 ijms-23-14141-f005:**
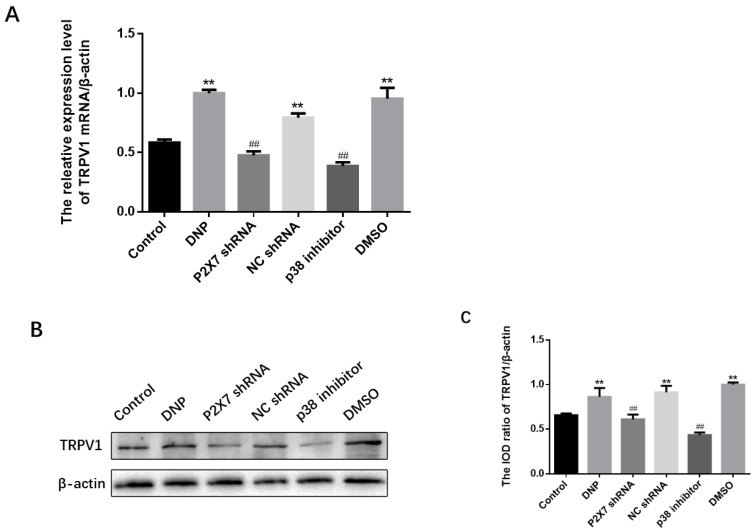
Measuring TRPV1 mRNA and protein levels and co-expression of NeuN and TRPV1 in DRG. (**A**) TRPV1 mRNA levels in the DRG. (**B**) TRPV1 protein bands in the DRG. (**C**) TRPV1 protein expression in DRG. (**D**) Co-expression of TRPV1 and NeuN in DRG. (**E**) Relative fluorescence intensity levels of NeuN and TRPV1 co-expression. TRPV1 mRNA, protein expression, and the co-expression of NeuN and TRPV1 in the P2X7 shRNA and p38 inhibitor groups were lower than in the DNP group. Blue staining (DAPI) marks the nucleus, green staining (NeuN) marks the neuron, red staining marks TRPV1, and the yellow signal is a combination of green and red signals. The scale bar represents 50 μm; data are shown as mean ± SEM based on at least three independent experiments, *n* = 6; ** *p* < 0.01 versus the Control group; ## *p* < 0.01 versus the DNP group.

**Figure 6 ijms-23-14141-f006:**
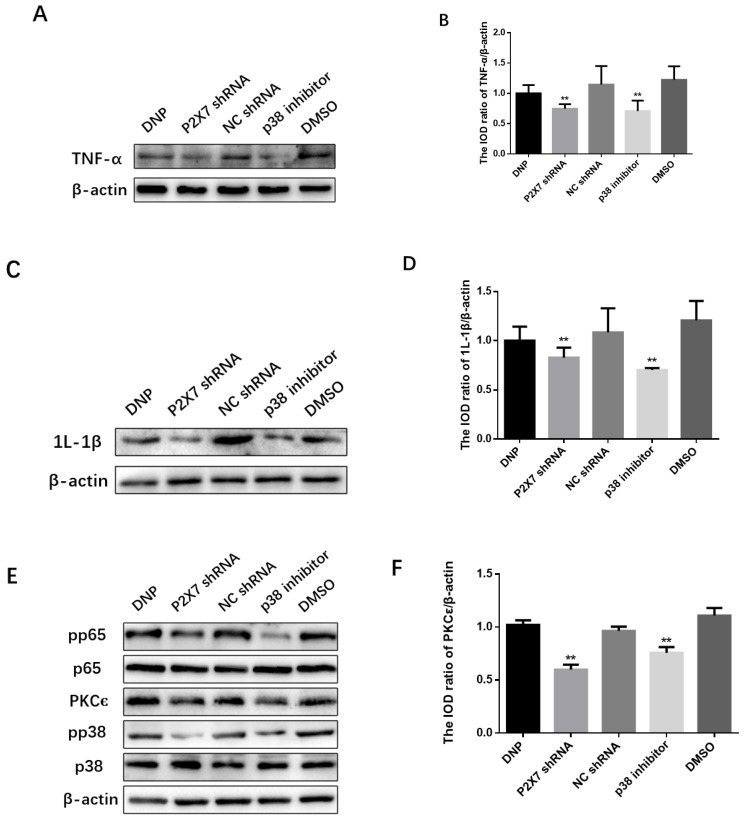
Expression of the PKCε/P38 MAPK/NF-κB pathway and inflammatory factors after activation. SDS-PAGE and relative protein expression of TNF-α (**A**,**B**) and IL-1β (**C**,**D**) in DRG. (**E**) SDS-PAGE of PKCε, p38, pp38, p65 and pp65 in DRG. Statistics of (**F**) PKCε/β-actin (**G**) pp38/p38 (**H**) pp65/p65; data are shown as mean ± SEM based on at least three independent experiments, *n* = 6; ** *p* ≤ 0.01 versus the DNP group.

## Data Availability

The original contributions presented in the study are included in the article. Further inquiries can be directed to the corresponding author.

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
