# Peer review of "Silencing P2X7R Alleviates Diabetic Neuropathic Pain Involving TRPV1 via PKCε/P38MAPK/NF-κB Signaling Pathway in Rats"

_ijms, 2022, doi:10.3390/ijms232214141_

Round 1

Reviewer 1 Report

The authors of this study investigated possible connection between the P2X7 receptor and TRPV1 in neuropathic pain in diabetic rats using a type 2 diabetic neuropathic pain rat model, intrathecal injection of P2X7 shRNA and p38 inhibitor, and recording of  abnormal mechanical and thermal pain and nociceptive hyperalgesia.  They also isolated dorsal root ganglia from the the spinal cord and measured pro-inflammatory cytokines, and examined the expression level of mRNA and protein for TRPV1, P2X7, IL-1β and  TNF-α .The results of these investigation showed that intrathecal injection of P2X7 shRNA and p38 inhibitor relieve hyperalgesia in diabetic  neuropathic pain rats. The aurthors suggest that glial P2X7 receptors are involved in diabetic neuropathic pain  by regulating neuronal TRPV1 expression via inflammatory factors TNF and 1Lbeta released from glia cells, and activating neuronal  PKCє/P38 MAPK/NF-κB pathway.

 These are important and  original results, but numerous details should be  included or explained, and several control experiments should be also shown.

 Major Comments:

The main problem of this study is that P2X7 and TRPV1 expression is documented on different cell types: the P2X7 was coexpressed with glia marker GFAP and TRPV1 with neuronal marker NeuN. Thus, P2X7 cannot regulate TRPV expression in the same cell, but inflammatory factors released from glia cells were considered as possible mediators of this effect. However, the release of inflammatory factors was not measured and activated glia cells could  release also other gliotransmitters such as ATP or glutamate via P2X7 pathway. Moreover, the PKCє/P38 MAPK/NF-κB pathway could be also active in glia cells. These possibilities are not discussed.

Paragraph describing chemicals used and their sources should be included into the section of Methods. Also sources and Companies providing the equipment  are not always given

Minor Comments:

 Line 2: ….intrathecal injection of   P2X7 shRNA and p38 inhibitor.

 Line 4: „As we know…“ References are missing in this sentence

 Line 44: What does it mean " good association" ? This association  should be explained in more details

 Line 49: „ P38 mitogen-activated protein….“ References are missing in this sentence

 Line 54: „PKC, MAPK signaling…….“ References are missing in this sentence

 Line 56: Citation # 13 is not original

 Line 58:  „…several physiological processes.“ , please specify these processes and include citations. Alternatively, remove this sentence

 Line 61: „upregulation of TRPV1“ is not documented in this study, because panels Fig.  5A, B and C do not show control values

Line 70:  Full name of ethic committee should be given

 Line 71: Rats of both sexes were used?

 Line 78: Abbreviation „NC“ is explained nowhere in the text

 Line 79:  p38 inhibitor is not specified

 Line 94: The time of injection is not mentioned. Also at the end of 5th week?

 Line 95: What is „SB203580“ ? This inhibitor should be mentioned earlier

 Line 120: “….acquisition system type RM40B“ Company and country should be given

Line 156:  What is  „RIPA lysate“?

 Line 244: „The dorsal rood nerve P2X7 receptor was co expressed with GFAP...“ Neurons do not express GFAP.

 Line 146: „...and the DNP groups (P>0.05).“ Please show results with quantification of immunofluorescence.

 Line 279: 6A

Line 282: 6B

Lines 307-308: „…the TRPV1 protein expression in the DNP group was higher than the  normal rats (Figure 5).“ Western blot and qPCR results do not show control groups in Fig.5, this statement is thus not documented 

 Line 308: Activation OR higher expression?

 Line  309-310:“The P2X7 receptor is a cation channel activated by high concentrations of…“ References are missing

 Lines 320-321: “Several studies…“ This sentence should be rephrased.

 Line 345:  „… we observed an increase in the number of satellite glial cells…“ This effect is not mentioned or quantified in Results section

 Lines 351-352:  „Glial cells interact ….„ Release of glia cells from glia cells is nonsense. This sentence should be rephrased

 Line355-356 : „DNP is the process…“This sentence should be rephrased.

 Figure 1: Describtion of X axis is missing. Time point of injection at the end of the fifth week  should be indicated in the figure

 Figures  2 and 3 are not mentioned in the text of Results

 Figure 4:  panels A, B and C are not mentioned within the text of Results. Panel D shows that the GFAP signal is  higher in all experimental groups as compared with  Control.  Please comment.

 Figure 5: Panels A, B and C are not mentioned within the text of Results. Control group is shown in only D panel,  please show controls also in A, B and C  panels. The DAPI and NeuN signal is  higher in some experimental groups as compared with Control, please comment.

 Figure 6 text: Panel A „a“ and „b „ show TNF, not 1Lbeta

Author Response

请参阅附件。

Reviewer 2 Report

Comment:

This paper discusses " Silencing P2X7R alleviates diabetic neuropathic pain involving 2 TRPV1 via PKCε/P38MAPK/NF-κB signaling pathway in rats ". The main contribution of the paper is P2X7 shRNA and p38 inhibitor relieved hyperalgesia in diabetic neuropathic pain rats and modulated inflammatory factors in vivo. P2X7 shRNA and P38 inhibitors significantly reduced TRPV1 expression by downregulating the PKCε/P38 MAPK/NF-κB signaling 23 pathway and inflammatory factors in dorsal root ganglia."

This is an interesting study and is generally well written and structured. However, in my opinion the paper has some shortcomings in regards to the recent references which are recommended.

Minor comments:

·         Well written except in some situations. I advise recheck it again.

·         The introduction should be advised to be re-written to be in more logical flow.

·         How many animals used for each experiments?

·         The methods in details should be described and analysis as well .

·         Please, Suggest future experiments in details

·         Please, Specify the more specific protein that you suggest might be related to this topic.

·         Please, try to add general paragraph about importance of this study. Why you studied this specific cells/specific model.

·         Figure 4 and 5 are not obvious to me. It is not understandable.

·         Figure 4 and 5 are too small.

·         Although it needs to be in more logical flow, the introduction provides a good, generalized background of the topic. However, why not cite more literature papers .

I suggest the followings:

1. Impairment in locomotor activity as an objective measure of pain and analgesia in a rat model of osteoarthritis.

https://www.spandidos-publications.com/10.3892/etm.2020.9294

2. The desensitization of the transient receptor potential vanilloid 1 by nonpungent agonists and its resensitization by bradykinin

https://journals.lww.com/neuroreport/Abstract/2020/08010/The_desensitization_of_the_transient_receptor.3.aspx

3. Analgesic Effects and Impairment in Locomotor Activity Induced by Cannabinoid/Opioid Combinations in Rat Models of Chronic Pain

https://www.mdpi.com/2076-3425/10/8/523

·         I think the motivations for this study need to be made clearer

·         Regarding the figures: I recommend make more figures to be illustrative.

Given these shortcomings the manuscript requires Minor revisions.

"I recommend that this paper be accepted after minor revision."

Author Response

请参阅附件。

Round 2

Reviewer 1 Report

The authors made numerous changes in the text that were not required and according to my opinion, were not necessary (the section of methods in the Abstract was rewritten uselessly, for example; practically all Introduction and a large portion of  Discussion was rewritten).  Consequently, resubmitted manuscript brought  new problems (abbreviation “STZ” for   “streptozocin” was not used in original version of manuscript,  and is not explained when used for the first time in revised version, see line 51, or in text to Fig. 1), and some parts of text disappeared (characterization of purinergic P2X7 receptor subtype, for example). In addition, the revised version of manuscript shows all changes that were made, and it is very difficult, practically impossible, to follow corrections that were required by reviewer. Thus, the present version represents rewritten text that needs reading of the whole manuscript again, which means practically new revision which is time consuming. In conclusion, the authors should submit revised version of manuscript with only required changes.  In addition, they should also submit   revised manuscript in version without marked changes, for better reading.  

Some problems with English persist. For example, the concluding sentence of the Abstract (lines 30 – 34) needs to be reformulated.

New references that have been added  are not highlighted in the List of references.

 Figures are shown 2 times, old  and new versions, but it is very confusing. In my opinion, the revised version should show only revised  figures.

I will wait for  new version where only required changes are made. Thus, few comments mentioned above are not final.

Author Response

请参阅附件

Round 3

Reviewer 1 Report

Line 59: instead of „Meanwhile, long[17] reported…“ should be „Meanwhile, Long et al. [17] have  reported…“

Line 163: ” 14000 x g”  , compare with Line 185: “ 12,000 ×g” . Writing of these numbers should be unified

Line 230: The authors stated in Methods that “Results were expressed as mean ± standard error of the mean, with P<0.05…“, but Figures 1 and 2 show also  P<0.01“,

Line 350: „We measured the protein expression of the expression level of TNF…“ this sentence should be rephrased

Lines 378-379: there is 2x “ in this study“,  this sentence should be rephrased

Lines 436-438: “ As we known, prolonged activation of P2X7R increases the release of inflammatory factors and the outflow of calcium ions into the extracellular environment[7]“. Concentration  of intracellular Ca is much lower compared to that in the extracellular environment and, consequently,  activated P2X7 channel mediates calcium influx, not release. This sentence should be corrected.
